# METAFS: AN EFFECTIVE WRAPPER FEATURE SELECTION VIA META LEARNING

## ABSTRACT

Feature selection is of great importance and applies in lots of fields, such as medical and commercial. Wrapper methods, directly comparing the performance of different feature combinations, are widely used in real-world applications. However, selecting effective features meets the following two main challenges: 1) feature combinations are distributed in a huge discrete space; and 2) efficient and precise combinations evaluation is hard. To tackle these challenges, we propose a novel deep meta-learning-based feature selection framework, termed MetaFS, containing a Feature Subset Sampler (FSS) and a Meta Feature Estimator (MetaFE), which transforms the discrete search space into continuous and adopts meta-learning technique for effective feature selection. Specifically, FSS parameterizes the distribution of discrete search space and applies gradient-based methods to optimize. MetaFE learns the representations of different feature combinations, and dynamically generates unique models without retraining for efficient and precise combination evaluation. We adopt a bi-level optimization strategy to optimize the MetaFS. After optimization, we evaluate multiple feature combinations sampled from the converged distribution (i.e., the condensed search space) and select the optimal one. Finally, we conduct extensive experiments on two datasets, illustrating the superiority of MetaFS over 7 state-of-the-art methods.

## 1 INTRODUCTION

Human activities and real-world applications generate a huge amount of data, where the data has a large number of features. Sometimes, we need to identify the effects of different features and select an optimal feature combination for further study. For example, on online shopping websites, marketer analyze the purchase data to study the purchasing preferences of different peoples and construct the users' profilingAllegue et al. (2020), which need to select a few number of representative shopping categories for further study. Supervised feature selection (FS) Cai et al. (2018b), making full use of label (e.g., the group classifications) information, is apposite to solve these problems.

Numerous supervised feature selection frameworks have been proposed, including filter Miao & Niu (2016), embedding Tibshirani (1996); Gui et al. (2019), and wrapper El Aboudi & Benhlima (2016) frameworks, shown in Figures 1(a, b, and c). Filter frameworks evaluate each feature individually while embedding frameworks greedily weighting features. They both evaluate features instead of combinations and the selection results are less effective. Wrapper frameworks traverse feature combinations and directly score them by wrapped supervised algorithms, *e.g.*, K-nearest neighbors (KNN) and Linear Regression (LR), which significantly improve the effectiveness of feature selection Sharma & Kaur (2021). However, the number of feature combinations grows exponentially with the number of features. Evaluating such large combinations is computationally intensive, and consequently a small number of evaluations would limit the improvement on the final performance. In summary, there are two major problems remaining in wrapper frameworks, which are listed as follows:

**Problem 1:** *How to search feature combinations in a large discrete space?* If we select $k$ effective features from a feature set with size $n$, we need to consider $\binom{n}{k}$ possible feature combinations, which forms an exponentially large search space. It's impractical to traverse and compare all feature combinations from a such huge discrete space.

Figure 1: Supervised Feature Selection Frameworks and our MetaFS

**Problem 2:** *How to efficiently and precisely evaluate feature combinations?* To evaluate subsets, we usually train an model from scratch for each subset, which is inefficient and time-consuming. To reduce the time consumption, researchers prefer to use some simple model such as KNN and LR as the evaluation model Rostami et al. (2021). However, these models are hard to capture the complex correlation among features and are hard to precisely evaluate different subsets (See Section 5.3).

To address these problems, as shown in Figure 1(d), we propose a meta-learning-based feature selection framework (MetaFS). MetaFS first transforms the discrete searching problem into a continuous optimization problem over a feature combination distribution, in which good feature combinations own higher sampling probabilities than bad ones. Specifically, we adopt differentiable parameters to represent the probability distribution function, such that it can be easily optimized. Next, we sample feature combinations from our distribution, and adopt deep meta-learning technique, which maintains the combination representations and generates unique evaluation model for each combinations without re-training, to solve the problem of efficient and precise feature combinations' evaluation. Then, according to the evaluation results, we increase the probabilities of good combinations and decrease bad ones by employing gradient-based method to optimizing the probability distribution function. Finally, we search the optimal combination in a condensed search space. The main contributions of our work are four folds:

- We propose a novel deep meta-learning-based feature selection framework, entitled MetaFS. It transforms the discrete search space into continuous and applies bi-level optimization, that alternatively condenses the continuous continuous search space and learns a meta-learning-based combination evaluation network for effective feature selection.

- We propose Feature Subset Sampler (FSS) to parameterize the distribution of search space, which allows us using gradient-based methods to condense the search space, effectively reducing the difficulties in the searching process.

- We propose Meta Feature Estimator (MetaFE) to efficiently and precisely evaluate feature combinations. It learns the representations of different feature combinations and generates unique evaluation model without re-training to each feature combination for precise evaluation.

- We conduct extensive experiments on two real-world benchmark datasets to verify our framework. The experimental results demonstrate the superiority of MetaFS.

## 2 RELATED WORK

**Feature Selection Methods** Cai et al. (2018b) can be classified into supervised, semi-supervised Yu et al. (2018); Liu et al. (2019) and unsupervised methods Liu et al. (2011); Tang et al. (2019). Unsupervised and semi-supervised methods cannot make full use of label information, having a poor performance on supervised tasks.

Filter frameworks, *e.g.,* MI Yang & Pedersen (1997), independently compute the correlation metrics of features. While embedding frameworks Gui et al. (2019); Singh et al. (2020) estimate the contribution (*e.g.,* attention and weight) of features in the process of model construction. Some neural-network-based embedding methods are proposed recent years. For example, Gui et al. (2019) carries out the feature selection in the learned latent representation space with an attention mechanism, Singh et al. (2020) selects features by an unique loss function and a well-designed neural network structure and Wang et al. (2019) iterative trains a hierarchical neural network for a better

selection. However, they all focus on feature scoring, but the greedily chosen features with high scores are usually not the optimal combination Wiegreffe & Pinter (2019); Jain & Wallace (2019).

Many works El Aboudi & Benhlima (2016) divided wrapper frameworks into heuristic-based methods Sharma & Kaur (2021); Rostami et al. (2021) and sequential selection methods Granitto et al. (2006). Sequential selections consecutively add (or eliminate) features from an empty (or full) set to select features. Whereas heuristic methods applies heuristic algorithms to search the best subset, such as particle swarm optimization (PSO) Karasu et al. (2020), genetic algorithms (GA) Sayed et al. (2019), and reinforcement learning algorithms Fan et al. (2020). However, these mentioned algorithms are still limited by the evaluation efficiency, hardly evaluating huge amounts of subsets during the selection procedure.

**Meta Learning for Weight Generation** Meta learning provides an alternative paradigm where a machine learning model gains experience over multiple learning episodes Hospedales et al. (2020); Vanschoren (2018). For example, Liu et al. (2018) learns the representation for different architecture and generate weights to accelerate architecture searching. And Li et al. (2019); Ha et al. (2016) generate weights for different data domain with the shared information. These mentioned meta learning model all focus on the similar datasets, architecture and hyper parameters, which aims to accelerate the convergence and improve the prediction accuracy by learning the shared representation among datasets. However, there is still no work for evaluating feature combinations.

## 3 PROBLEM FORMULATION

Suppose there are $n$ samples and $m$ features. Let $\mathbf{X} = [\boldsymbol{x}_1, \cdots \boldsymbol{x}_m]^\top \in \mathbb{R}^{n \times m}$ denote the input features, where each $\boldsymbol{x}_k \in \mathbb{R}^n$ is the $k$-th feature in input. Let $\boldsymbol{y} = [y_1, \cdots y_n] \in \mathbb{R}^n$ denote labels. Our goal is to find a feature subset $\mathbf{I} = \{i_1, \cdots, i_r\}$ with its corresponding input data $\mathbf{X}_{\mathbf{I}} = [\boldsymbol{x}_{i_1}, \cdots \boldsymbol{x}_{i_r}]^\top \in \mathbb{R}^{n \times r}$, such that we can achieve the minimum loss on the validation dataset by using the model trained with the training dataset. The optimization target can be formulated as:

$$\min \sum_{\mathbf{I} \in \mathbb{I}} \lambda_{\mathbf{I}} \mathcal{L}\left(\mathbf{X}_{\mathbf{I}}^{\mathrm{val}}, \boldsymbol{y}^{\mathrm{val}}; \Theta_{\mathbf{I}}\right), \quad \text{s.t.} \begin{cases} \Theta_{\mathbf{I}} = \arg\min_{\Theta^\star} \mathcal{L}\left(\mathbf{X}_{\mathbf{I}}^{\mathrm{train}}, \boldsymbol{y}^{\mathrm{train}}; \Theta^\star\right) \\ \sum_{\mathbf{I} \in \mathbb{I}} \lambda_{\mathbf{I}} = 1, \forall \lambda_{\mathbf{I}} \in \{0, 1\} \end{cases} \quad (1)$$

where $\mathbb{I}$ represent the set of all subsets, $\lambda_{\mathbf{I}}$ is a binary variable denotes whether subset $\mathbf{I}$ is selected or not, $\Theta_{\mathbf{I}}$ denotes the trainable parameters for subset $\mathbf{I}$, and $\mathcal{L}$ denotes the loss function.

## 4 METHODOLOGIES

As there are exponential numbers of subsets and $\lambda_{\mathbf{I}}$ are discrete, Eq. 1 is hard to be optimized. Inspired by Lovász (1975), we **relax $\lambda_{\mathbf{I}}$ to be continuous**, *i.e.,* $0 \leq \lambda_{\mathbf{I}} \leq 1$. In this way, the subset with the maximum $\lambda_{\mathbf{I}}$ is equivalent to the optimal result in Eq. 1 and we can apply gradient-based method to continuously optimize the feature selection.

In this paper, we propose MetaFS, consisting of two components: (1) *Feature Subset Sampler* (FSS) and (2) *Meta Feature Estimator* (MetaFE). Specifically, FSS parameterizes a subset sampling distribution to model the continuous coefficients $\lambda_{\mathbf{I}}$ and samples subsets for MetaFE. Next, MetaFE, applying meta learning techniques, efficiently generates unique models without re-training to precisely evaluate sampled subsets. Then, FSS and MetaFE are collaboratively optimized to condense the search space (increase the sampling probabilities of good subsets) and finally find the optimal one. In the following subsections, we'll show the structures of FSS, MetaFE, and the collaborative optimization process.

### 4.1 FEATURE SUBSET SAMPLER

There are $\binom{m}{r}$ coefficients $\lambda_{\mathbf{I}}$ in total, which is too large and cannot be directly maintained. As the constraint of $\lambda_{\mathbf{I}}$ (after relaxation) is similar to that of a probability distribution, we propose Feature Subset Sampler (FSS), using a probability distribution to model these coefficients:

$$\lambda_{\mathbf{I}} \leftarrow \mathcal{P}_{\mathrm{set}}(\mathbf{I}), \mathbf{I} = \{i_1, \ldots, i_r\} \in \mathbb{N}^r \quad (2)$$

where $\mathcal{P}_{\text{set}}(\mathbf{I})$ is the sampling probability of subset $\mathbf{I}$. For each sampling, we select $r$ features according to features' selecting probabilities without replacement. Suppose $p_i$ is the probability for the $i$-th feature to be select in a single sampling, the subset sampling probabilities are formulated as:

$$\mathcal{P}_{\text{set}}(\mathbf{I}_i) = \frac{\prod_{i_j \in \mathbf{I}_i} p_{i_j}}{\sum_{\mathbf{I}_k \in \mathbb{I}} \prod_{i_j \in \mathbf{I}_k} p_{i_j}}, \tag{3}$$

where $\mathbb{I}$ represent the set of all possible subsets. And when the features' selecting probabilities is fixed, $e.g.$, sampling subsets, the denominator is a constant and the subset sampling probability is proportional to the product of the features' selecting probabilities, $i.e.$, $\mathcal{P}_{\text{set}}(\mathbf{I}_i) \propto \prod_{i_j \in \mathbf{I}_i} p_{i_j}$.

FSS maintains $m$ trainable features' importance $\boldsymbol{c} = [c_1, \ldots, c_m] \in \mathbb{R}^m$ to model the feature selected probability $[p_1, \ldots, p_m]$. In this paper, we employ softmax function and define the selected probability of feature $i$ is:

$$p_i = \frac{\exp(c_i)}{\sum_{k=1}^m \exp(c_k)}. \tag{4}$$

Then, the sampling probability of subset $\mathbf{I}_i$ is proportional to the exponent of the sum of corresponding features' importance:

$$\mathcal{P}_{\text{set}}(\mathbf{I}_i) \propto \prod_{j \in \mathbf{I}_i} p_j \propto \prod_{j \in \mathbf{I}_i} \exp(c_j) = \exp \sum_{j \in \mathbf{I}_i} c_j \tag{5}$$

In the training process, FSS calculates the feature selecting probability $p_k$ according to the maintained feature importance $\boldsymbol{c}$ and samples subsets without replacement for MetaFE. Then, MetaFE calculates the evaluation of sampled subsets and feed backward for FSS to increase the sampiling probabilities of good subsets. (See detailed training stage in Section 4.3)

## 4.2 META FEATURE ESTIMATOR

Conventionally, when we evaluate different subsets, we need to train a unique model from scratch for each of them, which is time-consuming. Intuitively, similar feature combinations take similar effects, which means that there are inherent correlations among the model parameters that learned from similar subsets. To make full use of such correlations, we propose Meta Feature Estimator (MetaFE), learning the representations of different feature subsets, to generate unique model parameters for each sampled subset without re-training, for efficient and precise subset evaluation.

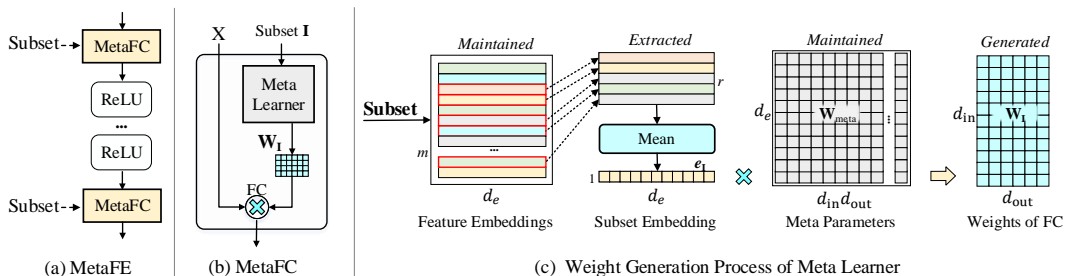

Figure 2: Meta Feature Estimator Framework

As shown in Figure 2(a), MetaFE is composed of activation functions (*e.g.,* ReLU and Sigmoid) and multiple Meta Fully Connected layers (MetaFC). Figure 2(b) shows the structure of MetaFC, which is a fully connected layer that calculates the output by generated parameters $\mathbf{W_I}$ that generated by Meta Learner for subset $\mathbf{I}$, formulating as:

$$\boldsymbol{x}_{\text{out}} = \boldsymbol{x}_{\text{in}} \mathbf{W_I} + \boldsymbol{b}, \mathbf{W_I} = \text{Learner}(\mathbf{I}) \tag{6}$$

Meta Learner maintains different subsets' representations by $m$ trainable feature embeddings. And it generates the weight parameters $\mathbf{W_I}$ according to the subset representation. As shown in Figure 2(c), let $[\boldsymbol{e}_1, \ldots, \boldsymbol{e}_m]^\top \in \mathbb{R}^{m \times d_e}$ denote $m$ trainable feature embeddings. For each input subset $\mathbf{I} = \{i_1, \ldots, i_r\}$, we suppose that all features have similar impact on model inference, meta learner first extracts the corresponding feature embeddings $\mathbf{E} = [\boldsymbol{e}_{i_1}, \ldots, \boldsymbol{e}_{i_r}]^\top \in \mathbb{R}^{r \times d_e}$ and fuses them

into a subset embedding $e_\mathbf{I} = (1/r) \sum_{k=i_1}^{i_r} e_k$ by mean function, as shown in Figure 2 (d). Then, meta learner generates the weight parameters $\mathbf{W_I} \in \mathbb{R}^{d_{\text{in}} \times d_{\text{out}}}$ by a trainable fully connected neural network (FC network) according to the learned subset embedding $e_\mathbf{I}$. And finally, we apply $\mathbf{W_I}$ to compute $x_{\text{out}}$ according to Eq. 6.

Note that instead of training FC weight parameters from scratch, MetaFE generates the parameters from subset embeddings and meta parameters (Figure 2b). In this way, MetaFE can generate unique fully connected neural networks for each input subset, and this generating process do not require re-training process, which can significantly improve the evaluation efficiency in feature selection.

## 4.3 SEARCHING PROCEDURE

Feature selection (Eq.1) can be regarded as a bi-level optimization problem Colson et al. (2007). Inspired by Cai et al. (2018a), we employ a collaborative optimization procedure to optimize MetaFE and FSS to finally search the best subset, which contains three stages and shown in Figure 3(a).

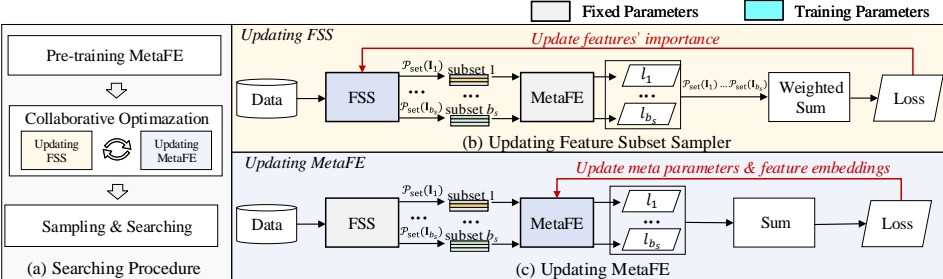

Figure 3: Training Process of Meta Feature Selection

**Pre-training MetaFE.** As MetaFE initially cannot compare the effectiveness of subsets without feeding any training data, we first pre-train MetaFE by using subsets sampled from uniform distribution. In this way, MetaFE could recognize the better feature combination to guide the optimization of FSS. The detailed training process of MetaFE is same to that in collaborative learning.

**Collaborative learning.** FSS and MetaFE are alternately optimized to condense the search space described by FSS and increase the evaluation accuracy of MetaFE, simultaneously:

- *Updating FSS.* FSS is trained to increase the sampling probability of good subsets and decrease the bad ones to condense the search space. As shown in Figure 3(b), we first sample a batch of subsets and evaluate them by MetaFE, and then calculate the gradient descent direction of features' importance to update FSS. Specifically, for a sampled subsets $[\mathbf{I}_1, \ldots \mathbf{I}_{b_s}]$ with the size number of $b_s$, we select a batch of samples for each of them, and feed these samples into MetaFE to calculate the supervised learning loss (subset evaluation) $[l_1, \ldots, l_{b_s}]$ of different subsets. Finally, we update the feature importance $c$ to increase the probability of good subsets, such that a subset with a lower loss need to have a higher sampling probability:

$$c^\star = \underset{c}{\arg\min} \sum_{k=1}^{b_s} l_k \mathcal{P}_{\text{set}}(\mathbf{I}_k) = \underset{c}{\arg\min} \sum_{k=1}^{b_s} l_k \exp \sum_{i \in \mathbf{I}_k} c_i, \quad (7)$$

where $\mathcal{P}_{\text{set}}(\mathbf{I}_k)$ is the sampling probability of subset $\mathbf{I}_k$ that is exponentially proportional to the sum of corresponding features' importance (Eq. 5). In this work, we employ gradient descent method to update the feature importance $c^\star$.

- *Updating MetaFE.* Similar to updating FSS, MetaFE is updated by batch gradient descent. For each update, we first samples $b_s$ subsets and select a batch of samples $[\mathbf{X}_{\mathbf{I}_1}, \ldots, \mathbf{X}_{\mathbf{I}_{b_s}}] \in \mathbb{R}^{b_s \times b_d \times r}$ for each of them, where $b_d$ is the batch size of samples. Then, we updates MetaFE to increase the accuracy of subsets' evaluation, such that,

$$\min_{\Theta_{\mathbf{I}_k}} \sum_{k=1}^{b_s} \mathcal{L}(\mathbf{X}_{\mathbf{I}_k}, y; \Theta_{\mathbf{I}_k}), \quad (8)$$

where $\mathcal{L}$ is the loss function, $\Theta_{\mathbf{I}_k} = \mathcal{G}(\mathbf{I}_k; \Theta)$ is the generated parameters that produced by subset $\mathbf{I}_k$ and all trainable parameters $\Theta$ in MetaFE (including subset embeddings and meta parameters). As a result, we can freeze the trainable parameters of FSS and optimize the trainable parameters $\Theta$ with the following formula:

$$\Theta^{\star} = \underset{\Theta}{\arg\min} \sum_{k=1}^{b_s} \mathcal{L}(\mathbf{X}_{\mathbf{I}_k}, \boldsymbol{y}; \mathcal{G}(\mathbf{I}_k; \Theta)). \qquad (9)$$

Then we can apply gradient descent to optimize all trainable parameters in MetaFE, and the gradient of $\Theta$ can be easily computed by back-propagation according to chaining rule.

**Sampling and Selecting.** when the search space is converged, MetaFS samples and evaluates multiple subsets. Finally, we select the optimal subset as our result.

## 5 EXPERIMENTS

In this section, we conduct experiments on two real-world datasets to show our superiority in three aspects: 1) the performance of the selected feature subsets; 2) the efficiency and precision of subset evaluation; 3) the convergence of subset search stages.

### 5.1 EXPERIMENTS SETTING

#### 5.1.1 DATASETS AND METRICS

The experiments are conducted on two datasets, one of them is public, listed as follows:

- *Shopping*: This dataset is from an online shopping website, which aims to estimate the customs' gender according to 21912 different shopping categories (features). This data is sparse and huge, which has more than 3 million samples. To reduce the time usage, we only select 10 thousand samples for baselines to evaluate each subset (use all samples in MetaFS).

- *Gisette* Dua & Graff (2017): Gisette is a handwritten digit recognition dataset, which has 7000 samples and 5000 features. And most of the features are highly correlated.

For all datasets, we simply select 50 features in experiments, so that we can artificially analyze them in downstream works. We divide all samples into two parts with a ratio of 8:2 for feature selection and testing. For the methods requiring validation dataset, we further divide the samples for feature selection into two parts, where 80% of them for training and 20% of them for validating. To test the performance of the selected subset, we extract the selected features and train a fully connected neural network (FCN) from scratch with an early-stop mechanism. And then we evaluate the subset performance on the test dataset. We also run such re-training processes 10 times, and take the mean values as the final results on Gisette dataset to reduce the evaluation bias as it has less samples and is easy to fall in over-fitting. We adopt precision, recall and F1-score to evaluate the results.

#### 5.1.2 BASELINES

We compare our MetaFS with 8 different methods, including four filter and embedding methods,

- *MI-KBest* Yang & Pedersen (1997): It ranks features by Mutual Information compared with label information. And then we select $k$ features with the highest scores.

- *Lasso* Tibshirani (1996): It trains a linear classification model (logistic regression) via $l1$ penalty, and select the features whose the corresponding weights are non-zero.

- *AFS* Gui et al. (2019): It trains a neural network and learns the feature importance with attention mechanism. We select the features that the corresponding attention weights are non-zero.

In addition, we compare MetaFS with four wrapper methods, where we apply Decision Tree (DT) as the wrapped model, as it has a good performance on subset evaluation (see Section 5.3).

- *PSO* Chuang et al. (2008): Particle Swarm Optimization (PSO) is a heuristic method, which optimizes a problem by iteratively trying to improve a candidate solution with regard to a given measure of quality. We use this method to search the best subset.

- *WOA* Mafarja & Mirjalili (2018): Similar to PSO, we apply whale Optimization Algorithm (WOA) to search the subsets, which mimics the hunting mechanism of humpback whales in nature.

- *HHO* Too et al. (2019): Hybrid High-Order (HHO) is another heuristic method, which searches subsets with several active and time-varying phases of exploration and exploitation.

We also compare a hybrid feature selection method, which is a mix of filter and wrapper methods:

- *PPIMBC* Hassan et al. (2021): It is a Markov Blanket based feature selection algorithm that selects a subset of features by considering their performance both individually as well as a group.

For all baselines, we turn the hyper-parameters and limit the numbers of selections closing to 50.

**Experiments settings** In our implementation, we build a MetaFE model with two hidden layers, where the layer sizes are set to $[64, 32]$. The size of feature embeddings in MetaFE is simply set to 256. The batch size of samples and subsets are set to 128 and 32, respectively. In the experiments, we train MetaFE on the training dataset, update FSS and search the optimal subset on the validation dataset. We employ cross entropy as a loss function in the experiments, and all experiments are conducted by using a single Nvidia 2070 GPU.

## 5.2 PERFORMANCE COMPARISON

Table 1 shows the evaluation results of the selected features. First, for the filter and embedding methods, *i.e.*, MI-Kbest, Lasso, and AFS, have worse performances on Gisette dataset, as they evaluate features greedily and are hard to capture the cross-correlation among features. Second, for the wrapper and hybrid methods, *i.e.*, RFE, PSO, WHO, HOO, and PPIMBC, which need to train plenty of models for evaluating different subsets, have bad performance on the Shopping dataset. The reason is that the Shopping dataset is hard to learn only with a few samples (see Section 5.3), and the methods easily fall into local minimum with imprecise evaluations.

Table 1: The Evaluation Performance of Selecting Results

| Metrics | Shopping | | | Gisette | | |
|---|---|---|---|---|---|---|
| | F1(%) | Precision(%) | Recall(%) | F1(%) | Precision(%) | Recall(%) |
| MI-Kbest | 0.7012 | 0.7045 | 0.7003 | 0.9456 | 0.9456 | 0.9456 |
| Lasso | 0.6625 | 0.6628 | 0.6624 | 0.9608 | 0.9607 | 0.9608 |
| AFSelect | 0.6870 | 0.6968 | 0.6865 | 0.9600 | 0.9599 | 0.9602 |
| PSO | 0.6782 | 0.6805 | 0.6775 | 0.9615 | 0.9615 | 0.9617 |
| WHO | 0.6721 | 0.6813 | 0.6718 | 0.9633 | 0.9632 | 0.9635 |
| HOO | 0.6256 | 0.6640 | 0.6336 | 0.9543 | 0.9543 | 0.9545 |
| PPIMBC | 0.4500 | 0.6136 | 0.5329 | 0.9725 | 0.9724 | **0.9728** |
| **MetaFS** | **0.7114** | **0.7151** | **0.7104** | **0.9726** | **0.9727** | 0.9725 |

Unlike the above methods, MetaFS achieves the best results on the two datasets. The reason is that MetaFS, as a wrapper framework, can directly compare different feature subsets during the selection procedure. In addition, compared to the other wrapper methods, MetaFS applies meta-learning techniques, efficiently evaluating subsets without re-training. Moreover, the deep-learning-based evaluation model can capture complex correlations among features, improving the evaluation's precision. With the precise and efficient subset evaluations, MetaFS could quickly filter out the bad subsets to condense the search space and finally tests much more subsets to select the optimal result.

## 5.3 STUDIES ON FEATURE SUBSET EVALUATION

To study the precision and efficiency of different evaluation methods, we sample 100 different subsets from the uniform distribution and a converged subset space. Then, we compare the time usage of the evaluation process and the ranking similarities of the evaluating results. The compared evaluation methods include four popular retrained learning methods, which are widely used in wrapper

methods, including KNN, Linear Regression (LR), Support Vector Classification (SVC), and Decision Tree (DT). In addition, we compare it with our proposed MetaFE. To illustrate the effectiveness of MetaFE, we also compared it to an FCN model that trained for all subsets without retraining.

Table 2: The Ranking Accuracy of Evaluation Results (RBO)

|  |  | Retraining | | | | Training Once | |
| --- | --- | --- | --- | --- | --- | --- | --- |
|  |  | KNN | DT | LR | SVC | FCN | **MetaFE** |
| Gisette | Random | 0.9092 | 0.9013 | 0.858233 | 0.8994 | 0.7190 | **0.9146** |
|  | Converaged | 0.6371 | 0.5747 | 0.6065 | 0.6313 | 0.5311 | **0.6772** |
| Shopping | Random | 0.6883 | 0.8061 | 0.7479 | 0.7600 | 0.6548 | **0.8518** |
|  | Converaged | 0.5106 | 0.5971 | 0.6470 | **0.6839** | 0.5265 | 0.6523 |

As shown in Table 2, we introduce *Rank Biased Overlap* (RBO) to evaluate the ranking similarities of different evaluation results, where a higher RBO indicates higher similarity. The ground truth is the results from an FCN model re-trained for each subset. For the Gisette dataset, as it is easy to learn, almost all re-trained methods have a good performance in the uniform sampling distribution. However, when the distribution converges, these models are hard to distinguish the nuances among good subsets precisely. For the sparse Shopping dataset, the re-trained methods produce worse results in all subset sampling spaces. The reason is that the efficiency of evaluation limits them (see Table 3), and they could only evaluate 10 thousand samples for each subset. Different from the above methods, MetaFE, based on a fully connected neural network, has an excellent ability to capture complex non-linear correlations among features. Moreover, it can almost ignore the time-consuming training stage and can be trained with all visible samples to improve the inference performance. Compared to FCN, MetaFE could generate unique model parameters for different subsets, which has a good generalization for subset evaluation. As a result, MetaFE has a precise evaluation performance in experiments, especially for datasets with a large number of samples.

Table 3: Time Usage of Evaluating One subset (ms)

|  | Retraining | | | | MetaFE | | |
| --- | --- | --- | --- | --- | --- | --- | --- |
|  | KNN | DT | LR | SVC | CPU | GPU | GPU* |
| Shopping(10k) | 7930.96 | 3715.88 | 3722.36 | 11572.44 | 375.85 | 31.26 | 73.6187 |
| Gisette | 533.51 | 35.65 | 33.76 | 366.34 | 27.92 | 7.24 | 13.905 |

Table 3 shows the time usage of different methods for evaluating one subset, where the re-trained methods only evaluate 10 thousand samples on the Shopping dataset. We also show the time usage of MetaFE under different devices, where the GPU* represents the estimated time usage that considers the training process. The re-trained methods cost plenty of time for training models. Worsely, the used time increases with the size of trained samples. Conversely, MetaFE does not need re-training models, costing less time. Based on a neural network, it can fully use GPU parallel technique, which significantly accelerates the calculation speed. Even though taking the training process into account (GPU*), the evaluation process is still much more efficient than re-trained methods. Therefore, MetaFE has excellent advantages in the precision and efficiency on subset evaluation.

## 5.4 CONVERGENCE ANALYSIS

In this subsection, we will analyze the convergence of MetaFS during the training process. We first compare the convergence curve of our MetaFS (FSS +MetaFE) with two variants to show the effectiveness of our training process: 1) FSS+FCN: replacing MetaFE with an FCN model, and 2) Init+MetaFE: removing the FSS and sample subsets from an initial uniform distribution.

We take the Gisette dataset as a typical case, and Figure 4 shows the F1-score varies over time. In the pre-training stage, all models are trained with random subsets (sampled from the uniform distribution). MetaFE can capture the inherent correlation of different subsets, having a higher prediction precision than FCN. In the collaborative learning stage, using FSS can increase the probabilities of good subsets and condense the search space. As a result, the estimator (MetaFE or FCN) can be trained with similar subsets and finally improves the precision of model inference. To show the change in model performance during training stages, we compare the difference of the subset evaluations of MetaFE that trained on different training stages. The ground truth is the test from a

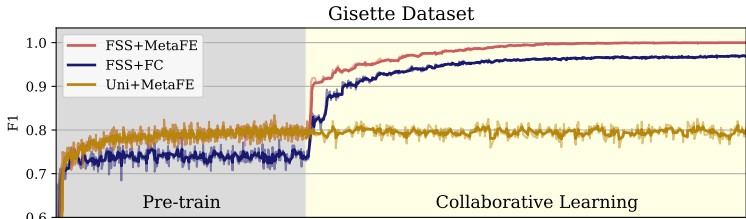

Figure 4: The Convergence Curves in Giseete Dataset

well-trained FCN model, and the subsets are sampled from two different distributions: 1) a uniform distribution and 2) a converged subset sampling space from a well-trained FSS.

Figure 5.4 shows the ranking similarity (RBO) change along the training iteration, where the blue lines are the results of MetaFEs that were trained on different training stages, where the evaluation subsets follow uniform distribution, corresponding to the RBO result on the left axis. The red lines are the results of MetaFEs evaluated with converged subset space, corresponding to the right axis.

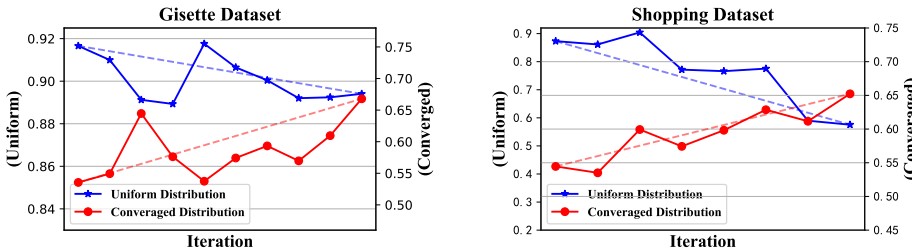

Figure 5: Ranking Similarity (RBO) of MetaFE Change along the Training Iteration

First, the RBO curves along the training iteration are unstable. One reason is that, since training FCN models for evaluating subsets is time-consuming, we only sample 100 subsets, which is small and cannot reflect the actual performance. Overall, the general trend of the RBO curve that evaluates converged subset space is increased with the training iteration. The reason is that MetaFE is collaboratively trained with the FSS and could specifically increase the precision of evaluating good subsets with the growing training iterations. For the subsets that follow a uniform distribution, the performance of MetaFE is decreased with the iteration as it is based on a neural network, having a characteristic of forgetting old knowledge Biesialska et al. (2020). However, our proposed collaborative training stage alternately optimizing the subset sampling space (FSS) and MetaFE, ensuring that MetaFE is always in the optimal state when updating the subset sampling space, can effectively avoid this problem. As a result, MetaFS has good performance throughout the training stages.

## 6    CONCLUSION AND FUTURE WORKS

In this paper, we proposed a novel Meta Feature Selection framework (MetaFS) for effective feature selection. MetaFS, consisting of FSS and MetaFE, transforms the discrete search problem into continuous and adopts meta learning techniques to evaluate subsets without retraining. Specifically, the proposed FSS parameterizes the search space and adopts gradient-based methods to effectively reduce the search difficulties. Meanwhile, ensuring the evaluation precision, MetaFE could significantly decrease the time usage of evaluation over the selection procedures. MetaFS adopts bi-level optimization strategies to optimize FSS and MetaFE, evaluating multiple subsets sampled from FSS and finally select the optimal one. We conduct extensive experiments on two datasets, demonstrating that our MetaFS searches the promising subsets. In the future, we will study the hyper parameters of MetaFS and simplify the operation to support rapid deployment in real-world scenarios.

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
