# OpenReview forum: "MetaFS: An Effective Wrapper Feature Selection via Meta Learning"
_ICLR.cc/2023/Conference — Submitted to ICLR 2023_

### Official Review · Reviewer_MBeM · 2022-10-22

**Confidence:** 4
**Correctness:** 3
**Technical Novelty And Significance:** 2
**Empirical Novelty And Significance:** 2
**Recommendation:** 3

**Clarity, Quality, Novelty And Reproducibility:**

Clarity: The manuscript has quite a number of typos. For example: "Converaged" , "Giseete dataset", missing x-axis label in Fig. 4, etc. Some notations are not defined properly. For example $\mathbf{I}_i$ was not defined before its first occurrence in Eq (3). It is also confusing because $i$ was also used to imply indices in the subset. Idea is overall clear, but background review could be more comprehensive.

Quality: Overall the methodology seems sound. I have a few questions/suggestions:

-- Can the author explain the formulation in Eq.(3)? I believe random sampling without replacement will result in a multivariate hypergeometric distribution, whose pmf doesn't look like what you described.

-- The author should consider including other prediction benchmarks without feature selection.

-- Please provide standard deviations for the reported results.

-- I am curious, what might have caused the sudden spike in performance right after pre-training?

-- Using two different y-axes for random and converged on the same plot (Fig. 5) is very misleading. Please use the same y-axis to plot these results. They are not that far apart in scales.

-- Since the proposed method has no theoretical result to back up its claim, I would generally expect more extensive empirical results. Demonstrating on two (heavily down-sampled) datasets, in my opinion, is quite insufficient.

Novelty: The approach is novel and quite intriguing.

Reproducibility: The code is provided. I did not try to run it, but I'm convinced that the results can be reproduced.




**Strength And Weaknesses:**

Strength: The approach is quite interesting and the result seems positive. The paper is clearly written.

Weakness:
1. The paper is not well motivated.
-- For example, the criticism for embedding methods is that "they all focus on feature scoring, but the greedily chosen features with high scores are usually not the optimal combination". I do not agree with this.

-- First, all neural networks are considered (embedding) feature extractors, and their features are learned optimally with backpropagation, which is not a greedy method at all.

-- Second, there is no guarantee that the proposed method selects optimal features either. As far as I understand, any subset selection method is just a special case of an embedding layer with output dim = input dim and zero weights given to unimportant features. What would be the value added of making these weights exactly zero?

2. The empirical result is not very convincing.
-- What is preventing us from learning an end to end neural network to predict given all features? 20000/5000 features are not prohibitively large numbers of features. Maybe such naive approach would perform worse than the proposed method in the end, but the authors should at least try to include such a simple baseline to demonstrate that these datasets truly need feature selection to perform well.

-- Why do we presume that the optimal subset has exactly 50 features? I think there should be a separate study conducted to verify this number. For example, will MetaFE still perform the best when the subset size is larger? Would there be scalability issues?

-- Evaluating the ranking accuracy is a fair approach to understand the optimality of the selected features. However, it seems that the ranking accuracy is quite bad on sample subsets drawn from the converged sampling distribution. I'm not fully convinced that the reason for this is necessarily "hard to distinguish among good subsets precisely". Perhaps one way to verify this exactly is to construct a synthetic dataset with many duplicated features and observe if MetaFE can recover one unique copy of each feature.

**Summary Of The Paper:**

This paper proposes a new approach for feature selection, which composes of two components: (1) A feature subset sampler, which learns the distribution of important feature subsets; and (2) a meta-learning based weight generator that quickly constructs a fully connected deep layer for any sampled subset. The method is demonstrated on two datasets. The empirical result is generally positive.

**Summary Of The Review:**

Overall, I would recommend a rejection. As stated above, given that this is an empirical work, I believe the reported result is insufficient. The motivation also failed to convince me that there is a practical need for the proposed method.

---

> ### Author Response · Authors · 2022-11-09
> **Reply to comments**
>
> Thank you for your advices.
>
> 1. Why we said "the embedding methods are all focus on feature scoring, but the greedily chosen features with high scores are usually not the optimal combination"?
>
>    The NN-based embedding methods usually evaluate features by weighting or attention mechanism. However, [1,2] have shown that the learned attention distribution is highly correlated with the rest of the layers, and the learned attention scores are not guaranteed to be the optimal solution. As a result, we said that the result selected by embedding methods is only a good solution for the embedded model but may not be the optimal solution for downstream tasks.
>
>    Conversely, wrapper methods usually evaluate subsets according to the scores evaluated by the downstream tasks (the wrapped model). Theoretically, they can select the optimal subset. However, limited by the time-consuming subset evaluation, existing wrapper methods still have much room for improvement. Our work is trying to avoid the re-training procedure in subset evaluation to reduce the time usage and improve the selecting result.
>
>    [1] Sarthak Jain and Byron C Wallace. Attention is not explanation. arXiv preprint arXiv:1902.10186,
>    2019.
>
>    [2] Sarah Wiegreffe and Yuval Pinter. Attention is not not explanation. arXiv preprint arXiv:1908.04626, 2019.
>
> 2. How does MetaFE work?
>
>    MetaFE is a meta-model. It generates different fully connected neural network parameters for different subset inputs to precisely evaluate subsets. The input dimension is the size of features $m$, and the output dimension is 2 (binary classification problems) in our experiments. MetaFE will set the corresponding data for each input to zero if the features are not in the given subset.
>
> 3. Why we select 50 features from 20000/5000 all features?
>
>    In one of our real application scenarios (Shopping dataset), researchers would manually analyze the selecting result of feature selection methods to ensure interpretability. For example, some criminals often have unusual shopping activities before the crimes, and researchers need to analyze what feature combinations are highly correlated with that. As a result, the selection size needs to be manually set and can be limited to a small size.
>
>    Unlike some other feature selection works, our model is more suitable for the case with large samples, as it can apply Batch Gradient Descent methods to utilize all data samples.
>
> 4. Will MetaFE still perform the best when the subset size is larger?
>
>    Yes. Actually, MetaFE will perform better with a larger subset size.
>
>    Note that MetaFE maintains lots of feature embeddings for all features. Suppose we select $r$ from $m$ features. For each training batch, the coverage rate of the network parameters (feature embeddings) is $r/m$. As a result, a larger subset size means a larger parameter coverage rate, making the neural network more stable.

---

> > ### Comment · Reviewer_MBeM · 2022-11-16
> > **Re: Point 1**
> >
> > Theoretically, NN-based embedding methods can select the right subset too. Consider a one-layer NN, we can always set the weights to a binary matrix such that it corresponds to a subset selector. The point is that there is no guarantee, just as there is no formal guarantee that MetaFS will select the optimal subset.

---

### Official Review · Reviewer_3nfV · 2022-10-22

**Confidence:** 4
**Correctness:** 3
**Technical Novelty And Significance:** 2
**Empirical Novelty And Significance:** 1
**Recommendation:** 3

**Clarity, Quality, Novelty And Reproducibility:**

The introduction and problem statement are clear, the paper does provide some novelty in the two components proposed by the authors.

The experimental part and results are not well described, and some details are hard to understand. This would make reproducibility limited.


**Strength And Weaknesses:**

The problem is well-studied but still important for the community. The English level is satisfactory. The method itself seems novel, and the authors propose a way to save computational time by exploiting feature correlation and training sharing weights for models that are trained on the same or similar features. The method is quite intuitive, and the continuous relaxation was already demonstrated effective by other works for feature selection, for example:
“Concrete Autoencoders” or “Feature Selection Using Stochastic Gates”.

Some of the sections in the paper do not read well and are hard to follow.
The experimental evaluation is quite poor, the authors only apply the method to two binary data sets (one of them is half synthetic- GISETTE). Also, several leading feature selection baselines are missing.  It is really hard to judge the quality of the approach without additional experiments. The runtime improvements seem promising, but an ablation study is missing from their experiments.


In the following, I detail my comments point by point:
P1) The authors mention in the introduction that they compare to SOTA in FS, this is not correct, several leading NN baselines are left out, for example:\\
[1] Lemhadri et al. Lassonet: Neural networks with feature sparsity
[2] Yamada et al. Feature Selection using stochastic gates, 2020
[3] Balin et al .Concrete autoencoders: Differentiable feature selection and reconstruction
[4] Singh et al. Fsnet: Feature selection network on high-dimensional biological data

P2) Also, experimenting with two datasets is not considered extensive.

P3) Wrong use of citation style, embed the names in the sentences or use brackets.

P4)Saying that the selection results of filter and embedding method is less effective is wrong and misleading the reader. There are many settings where filter methods excel and save computational time. Furthermore, embedding methods can work with strong NN models and lead to SOTA results.
Conceptually wrapper methods can outperform them, but this might require an unfeasible amount of training time.

P5)Problem 2: an model-> a model

P6) GISETTE is a semi-synthetic data created for the NEURIP 2003 challenge, with most of the features being a nuisance, created artificially. The authors don’t even mention this.

P7) Why do you only select 10K samples for baselines in the Shopping data? And use the full data for your method? This seems unfair.


P8) We only select 50 features in experiments….??? What does this sentence mean?


P9) You evaluate precision-recall and F1, but you don’t mention what? Namely does can also be evaluated in terms of the ability to identify informative features in GISSETTE (where the set is known). I assume you mean for binary classification, so what threshold do you use to set precision-recall?

P10) We turn the hyperparameters? Do you mean tune? Also, how do you tune them for all baselines? This is not explained.

P11) The network used is quite narrow and shallow, nowadays theory and practice indicate we need to use wider and deeper models.

P12) Section 5.3 is not clear. For example, “we sample 100 different subsets..” of what? I can try to guess what this means, but a paper should be clearly written for readers.

Strong NN FS methods are missing in the related work and evaluations [1]-[4] above



**Summary Of The Paper:**

The authors address a challenging and important problem of feature selection. The new approach is a wrapper-type method that relies on meta-learning to select the best subset of features for supervised learning. The idea is to transform the large discrete search space into a relaxed continuous space and use gradient descent to find the optimal subset for classification/regression.

**Summary Of The Review:**

Overall the authors present a new method for an important problem. They present two components that should reduce the computational effort when using wrapper methods for FS. However, a large portion of the paper is not written in a clear way, and the experiments only focus on two binary datasets. One of them is semi-synthetic. Based on such a small evaluation, I can’t recommend accepting the paper.

---

> ### Author Response · Authors · 2022-11-09
> **Replay to comments**
>
> Thank you for your advices.
>
> 1. There are some NN-based baselines that are not compared.
>
>    The proposed MetaFS is a wrapper-based feature selection method, and we mainly compared it with the other wrapper-based methods. The mentioned NN-based baselines [1-4] are all embedding-based, which evaluate features instead of feature subsets. Generally, these methods are fast but less effective. As a result, we didn't compare too many NN-based baselines.
>
> 2. Saying that the selection results of filter and embedding method is less effective is wrong and misleading the reader.
>
>    In theory, filter and embedding methods greedily evaluate features instead of feature subsets, which have a fast speed and may perform well. While the wrapper methods directly evaluate feature subsets, which theoretically have a better result. But the evaluation speed limits the performance. Therefore, our work tries to decrease the time usage in subset evaluation to close the result to the theoretical upper limit.
>
> 3. Why use 10K samples for baselines in the Shopping data? This seems unfair.
>
>    For the filter and embedding baselines, e.g., MI-Kbest, Lasso, and PPIMBC,  the existing implementations are to load all the data into memory and then calculate the results.  Using all data to train the model will run out of memory (approximately 260GB). The other wrapper baselines, e.g., PSO, WOA, and HHO, need to re-train a model to get its precise evaluation. While the time spent increases with the number of data samples. Conversely, our method can apply Batch Gradient Descent methods to utilize all data samples.  (Although the AFS baseline is NN-based, it utilizes the result of Kbest for model pre-training, and it is hard to use all of the samples.)
>
> 4. The network used is quite narrow and shallow.
>
>    The insight of our MetaFE is to generate a unique neural network model for each subset. For the generate model (here, we use a fully connected neural network structure), as the number of selected features is small, a giant model easily falls into over-fitting.
>
>    For the Meta model (MetaFE), We had tried different structures to generate the evaluation models, including some attention mechanisms. However, the experiments show that complex structures don't work well.
>
> 5. What does the sentence "We only select 50 features in experiments" mean? And how do you tune the hyper-parameters for all baselines to limit the number of selection?
>
>    We select 50 features to analyze the performance of feature selection methods from all features. For the methods that could dynamically change the number of selections, we also tune the hyper-parameters and limit the number of selection closing to 50 (no less than 50). Even some baselines are had to control. For example, PPIMBC selected 400+ features.
>
>    For MI-Kbest, we can directly set the number of selections. For the other baselines, e.g., AFS, PPIMBC, and the other wrapper methods, we change the hyper-parameter that controls the masks.

---

> > ### Comment · Reviewer_3nfV · 2022-11-21
> > **After rebuttal**
> >
> > I want to thank the authors for replying to my comments.  I think that the authors have presented an interesting idea; unfortunately, I still feel that the paper needs substantial improvement before publication.

---

### Official Review · Reviewer_z1i8 · 2022-10-29

**Confidence:** 2
**Correctness:** 3
**Technical Novelty And Significance:** 3
**Empirical Novelty And Significance:** 3
**Recommendation:** 6

**Clarity, Quality, Novelty And Reproducibility:**

Clarity, quality, and novelty of the paper is satisfactory. They also provide the code

**Strength And Weaknesses:**

Strength
- Paper is well written and easy to understand
- The problem is important and applying meta-learning to solve the problem seems appropriate.
- The empirical results are favorable, covering wide variety of existing baselines.

Weaknesses
- The authors only use two datasets for evaluation, which is quite limiting. I suggest the authors to use more datasets to properly assess the efficacy of their method.
- The gap from the baselines are quite marginal, and all the experimental evaluation lacks confidence interval. So one cannot say that the evaluation results are statistically significant.

**Summary Of The Paper:**

This paper tackles feature selection problem by using meta-learning. They propose to transform the discrete search problem into continuous one and make use of meta learning to evaluate subsets without retraining. They parameterize the search space and apply gradient-based meta-learning in order to effectively reduce the search difficulties. The experimental results show that the proposed method outperforms many of the existing feature selection methodologies.

**Summary Of The Review:**

In summary, I think the paper tackles the important problem of feature selection, also the method seems appealing as well. However, evaluations are a bit limited, so I encourage the authors to evaluate on more datasets and run all the baselines multiple times to judge the statistical significance.

Overall, I think the strength outweigh the weaknesses, so I recommend weak accept.

---

> ### Author Response · Authors · 2022-11-09
> **Reply to review comments**
>
> Thank you for your advice. We'll add more datasets in the future.
>
> The reason for choosing these two datasets is to show the performance of the methods applied in low and high numbers of samples. And the results have shown the superiority of our methods.
>
> We did not focus more on the selections' performance. One of our goals is to find a way to efficiently and precisely evaluate feature subsets for wrapper-based feature selection. And the improvement of the final result (in Section 5.2) is due to the larger amounts of searches compared with the other methods. Based on this reason, though the gap from the baselines is small, it has shown our methods' effectiveness. In more detail, we compared the efficiency and precision of several evaluation methods in Section 5.3.

---

### Official Review · Reviewer_joqh · 2022-11-04

**Confidence:** 5
**Correctness:** 3
**Technical Novelty And Significance:** 3
**Empirical Novelty And Significance:** Not applicable
**Recommendation:** 5

**Clarity, Quality, Novelty And Reproducibility:**

The logic is clear, but the presentation is not well-polished. Some ideas are novel, but the framework has big flaws. There seems no big issue with reproducibility.

**Strength And Weaknesses:**

Strengths:
1. The research problem is important and the idea of MetaFE is meaningful. Wrapper is one of the most widely used strategies in feature selection problems. The major challenge is the repeated and redundant training of the machine learning models. The proposed MetaFE tackle the re-training issue with meta-learning. It could be especially useful for big data.
2. The FSS transforms an exponential combinatorial optimization problem into a continuous optimization problem, where most existing optimizers in deep learning can be directly adopted. This provides a new perspective for combinatorial optimization.

Weaknesses:
1. The manuscript is not well polished. There exist many grammar errors, e.g., 'study the purchasing preferences of different peoples', 'construct the users’ profilingAllegue et al.', 'employing gradient-based method to optimizing the probability distribution'.
2. The correlation between features is ignored, which is one of the most important factors in feature selection problems. Essentially, in this paper, each feature is independently quantified by an importance number $c$. Feature selection needs a joint distribution of all features, instead of one distribution for each feature.
3. The proposed method can not decide the number of selected features. Besides, there is a lack of analysis of the selected feature number.

**Summary Of The Paper:**

This paper proposes a meta-learning-based feature selection framework that aims to select effective features for machine learning tasks. Specifically, this wrapper framework consists of two components, i.e., Feature Subset Sampler (FSS) which optimizes the discrete search space in a continuous way, and Meta Feature Estimator (MetaFE) which alleviates the cost of re-training machine learning models.

**Summary Of The Review:**

This paper proposed a wrapper framework to solve feature selection problems. The idea of Feature Subset Sampler (FSS) optimizes the discrete space in a continuous way, and the Meta Feature Estimator (MetaFE) avoids the re-training problem. However, the paper is not well polished, the framework ignores feature-feature correlation, and it lacks solutions to deciding selection feature numbers.

---

> ### Author Response · Authors · 2022-11-09
> **Reply to review comments**
>
> Thank you for your advice. We will fix the grammar errors in this paper.
>
> QA:
>
> 1. MetaFE quantifies the features only by an importance number c, which may ignore the correlation among different features.
>
>    MetaFE maintains $n$ different $d_e$-dimensional vectors to represent the features. The vector not only represents the feature's importance but also represents how the feature works in a subset.
>
>     MetaFE will generate the subset embedding from the corresponding vectors in the training process. Here, we fuse the vectors by a mean function to learn the correlation among features. Actually, we had tried to use some learning structure or an attention mechanism to generate the subset embedding. However, the experiments show that complex structures don't work well.
>
> 2. The proposed method can not decide the number of selected features, and there is a lack of analysis of the selected feature number.
>
>    In one of our real application scenarios, researchers would manually analyze the selecting result of feature selection methods to ensure interpretability. For example, some criminals often have unusual shopping activities before the crimes, and researchers need to analyze what feature combinations are highly correlated with that. As a result, they prefer to set the number of selections manually.
>
>    In our experiments, We select 50 features to analyze the performance of feature selection methods. For the methods that could dynamically change the number of selections, we also tune the hyper-parameters and limit the number of selection closing to 50 (no less than 50).

---

### Decision · Program_Chairs · 2023-01-20

**Decision:**

Reject

**Justification For Why Not Higher Score:**

The presentation and experimental evaluations should be much improved for future submissions.

**Justification For Why Not Lower Score:**

N/A

**Metareview: Summary, Strengths And Weaknesses:**

This paper leverages a meta-learning scheme to over two challenges present in a wrapper feature selection method. The idea seems to be interesting and the problem itself is important. However, there are critical concerns that were not resolved. Some reviewers pointed out that some of sections are not easy to read. Thus, I would like to suggest the authors to improve the presentation for future submissions. Another weakness in this paper lies in experimental evaluations. Most of reviewers criticized that experimental evaluations should be dramatically improved, with more baselines and datasets. The results are very marginal over existing methods. Therefore, the paper is not recommended for acceptance in its current form. I hope authors found the review comments informative and can improve their paper by addressing these carefully in future submissions.